# Multi-Attribute Constraint Satisfaction via Language Model Rewriting

**Ashutosh Baheti**$^{\diamond,\clubsuit,*}$**, Debanjana Chakraborty**$^{\blacklozenge}$**, Faeze Brahman**$^{\clubsuit}$**, Ronan Le Bras**$^{\clubsuit}$**,**
**Ximing Lu**$^{\heartsuit,\clubsuit}$**, Nouha Dziri**$^{\clubsuit}$**, Yejin Choi**$^{\heartsuit,\clubsuit}$**, Mark Riedl**$^{\diamond}$**, Maarten Sap**$^{\spadesuit,\clubsuit}$
$^{\diamond}$ Georgia Institute of Technology, $^{\heartsuit}$University of Washington, $^{\blacklozenge}$The Ohio State University,
$^{\spadesuit}$Carnegie Mellon University, $^{\clubsuit}$Allen Institute for Artificial Intelligence
$^{*}$`abaheti95@gatech.edu`

Reviewed on OpenReview: `https://openreview.net/forum?id=3q1bUIHTJK`

## Abstract

Obeying precise constraints on top of multiple external attributes is a common computational problem underlying seemingly different domains, from controlled text generation to protein engineering. Existing language model (LM) controllability methods for multi-attribute constraint satisfaction often rely on specialized architectures or gradient-based classifiers, limiting their flexibility to work with arbitrary black-box evaluators and pretrained models. Current general-purpose large language models, while capable, cannot achieve fine-grained multi-attribute control over external attributes. Thus, we create Multi-Attribute Constraint Satisfaction (MACS), a generalized method capable of finetuning language models on any sequential domain to satisfy user-specified constraints on multiple external real-value attributes. Our method trains LMs as editors by sampling diverse multi-attribute edit pairs from an initial set of paraphrased outputs. During inference, LM iteratively improves upon its previous solution to satisfy constraints for all attributes by leveraging our designed constraint satisfaction reward. We additionally experiment with reward-weighted behavior cloning to further improve the constraint satisfaction rate of LMs. To evaluate our approach, we present a new Fine-grained Constraint Satisfaction (FINECS) benchmark, featuring two challenging tasks: (1) Text Style Transfer, where the goal is to simultaneously modify the sentiment and complexity of reviews, and (2) Protein Design, focusing on modulating fluorescence and stability of Green Fluorescent Proteins (GFP). Our empirical results show that MACS achieves the highest threshold satisfaction in both FINECS tasks, outperforming strong domain-specific baselines. Our work opens new avenues for generalized and real-value multi-attribute control, with implications for diverse applications spanning natural language processing and bioinformatics.

## 1 Introduction

Multi-attribute constraint satisfaction is a challenging problem that holds many useful applications in the domains of natural language processing (NLP), drug design, and protein engineering. In NLP, numerous classifiers and regressors exist for detecting individual linguistic attributes such as fluency, sentiment, formality, and complexity. Enabling *fine-grained granular control* over such attributes will allow users to personalize any text with their desired style (Kumar et al., 2021; 2022). In the realm of medicine and biotechnology, fine-grained control of multiple physicochemical properties opens avenues for engineering of novel drugs and proteins, for example, antibiotics with increased efficacy and reduced toxicity (Wong et al., 2023), and specialized proteins with manipulated attributes like fluorescence, binding affinity (Shen et al., 2014), and stability (Chan et al., 2021).

Conventional methods for multi-attribute control often rely on mechanisms such as class-conditioned LMs (Keskar et al., 2019; Lu et al., 2022; Hallinan et al., 2023) or latent attribute embeddings (He et al., 2020; Russo

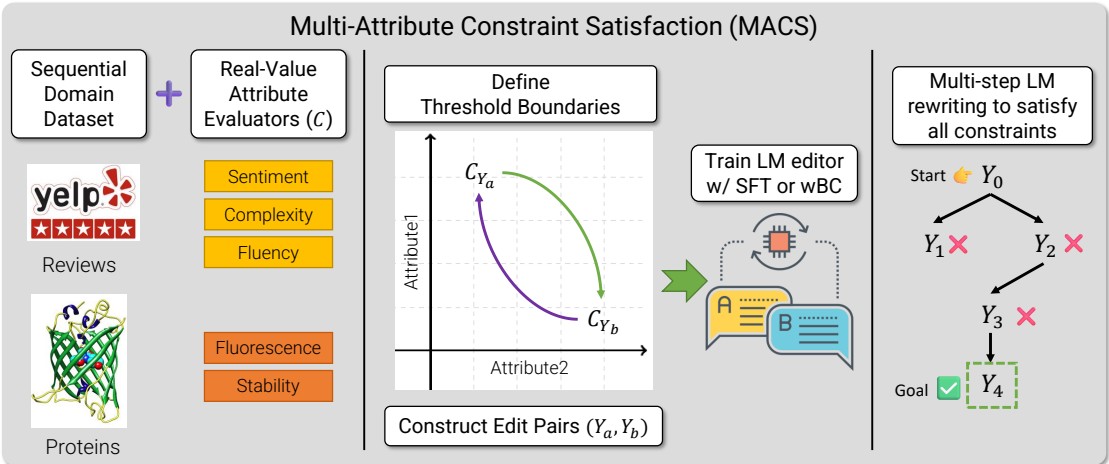

Figure 1: MACS framework starts with sequential domain datasets (customer reviews or proteins) and a set of real-value attribute evaluators (such as sentiment, complexity regressors, or protein folding models). We then define fine-grained threshold-window boundaries for every attribute and create edit pairs distributed across the multi-attribute landscape. We train the LM editor on top of the edit pairs by leveraging supervised fine-tuning (SFT) or reward-weighted behavior cloning (wBC). Subsequently, LM editors can achieve the desired fine-grained constraints by employing prioritized editing that maintains a priority queue of past edits ordered by their proximity to the target threshold constraints.

et al., 2020; Riley et al., 2021; Gu et al., 2022; Liu et al., 2022; Ding et al., 2023). However, these approaches are generally proposed to handle attributes with categorical values and may not effectively generalize to those with continuous/scalar values, such as gradually changing the complexity/readability of a sentence or modifying the activity of a protein within specified bounds. Techniques that do support real-value constraints satisfaction often require computationally expensive gradient-based decoding optimizations (Dathathri et al., 2020; Kumar et al., 2021; 2022; Qin et al., 2022; Li et al., 2022) or energy-based sampling (Mireshghallah et al., 2022; Liu et al., 2023). These methods suffer from slow inference and fixed-length outputs, limiting their widespread adaption to downstream applications.

We introduce the **Multi-Attribute Constraint Satisfaction (MACS) framework**, a generalized method for training LMs from diverse sequential domains towards fine-grained constraint satisfaction. Here, the LM is conceptualized as an **editor** tasked with navigating the multi-attribute landscape, iteratively refining its outputs to meet desired constraints. Unlike online and off-policy reinforcement learning (RL) methods (Lu et al., 2022; Hallinan et al., 2023), which require LM editor-generated data during training, our approach utilizes an initial set of paraphrased outputs and directly trains the LM editor on them by sampling edit pairs. The initial paraphrases are obtained externally. In the case of text style transfer, for example, through few-shot prompting in the language domain. In the case of protein design, through randomized mutations in the protein domain. As part of the framework, we introduce a generalized reward function for constraint satisfaction and experiment with offline reward-weighted behavior cloning (Norouzi et al., 2016) to train the fine-grained LM editor on sampled edit pairs. Finally, we introduce a reward-prioritized multi-step inference strategy to enhance constraint satisfaction while reducing overall inference computational cost. We provide an overview of our learning and evaluation process of MACS in Figure 1.

To comprehensively assess the effectiveness of MACS, we introduce a novel Fine-grained Constraint Satisfaction (FINECS) benchmark, that consists of two challenging controllability tasks. The first task, *Text Style Transfer* (§4), requires precisely modifying the sentiment and complexity of a given text while preserving its fluency and content similarity. We sub-divide sentiment and complexity attribute ranges into five threshold window constraints each, resulting in 25 different multi-attribute threshold combinations. The second task, *Protein Design* (§5), focuses on mutating the Green Fluorescent Protein (GFP) to achieve the desired fluorescence and stability, with both attributes divided into four threshold windows, leading to a total of

16 threshold-window combinations. During evaluation, the LM editors are tasked with satisfying every multi-attribute constraint within a fixed inference budget. We analyze the trade-off between different input conditions, inference strategies, and objective functions.

A systematic comparison of our framework against preexisting domain-specific baselines[1] shows that LM editors trained with MACS yield the highest constraint satisfaction success rate in both FINECS tasks. Our findings show the potential of adapting language models from diverse domains as fine-grained editors that allow navigating across the multiple-attribute landscape and discovering novel sequences. We release the code at `https://github.com/abaheti95/MACS`.

## 2 Related Work

**Precise Multi-Attribute Control**   While controlled text generation and style transfer have been widely studied problems in NLP literature, enabling fine-grained constraint satisfaction still proves to be quite challenging. A prominent approach is to incorporate attribute signals in gradients during decoding to allow satisfying multiple attribute constraints on them (Dathathri et al., 2020; Kumar et al., 2021; 2022; Qin et al., 2022; Li et al., 2022; Liu et al., 2023). However, there are three major limitations of these methods, (1) they require white-box access to evaluators for gradient computation, (2) their decoding speed is slow and memory intensive, and (3) their output length needs to be predefined to tractably compute the gradients. Other studies have proposed architecture augmentations and specialized loss functions (Russo et al., 2020; Riley et al., 2021; Gu et al., 2022; Liu et al., 2022; Ding et al., 2023; Hu et al., 2023) to perform multi-attribute control of language models. However, they typically cannot work with arbitrary external attribute evaluators and some also require expensive on-policy or off-policy samples during training. Recently, Mireshghallah et al. (2022) proposed probabilistic energy models to allow black-box attribute scorer constraints, but can only use masked language models for output sampling. In contrast to all the above methods, our framework leverages offline learning to offer the most flexibility in terms of external scorers, LM architecture choice, and training data sources.

**Iterative Refinement via verbal feedback**   LLMs may not generate the best output on their first attempt. Therefore, many recent prompting methods have been introduced for LLMs to iteratively improve model outputs while incorporating internal and/or external LLM evaluators as verbal feedback (Shinn et al., 2023; Zhang et al., 2023; Madaan et al., 2023; Dhuliawala et al., 2023; Akyurek et al., 2023; Gou et al., 2024). However, these methods implicitly expect the availability of expert large language model (LLM) which may become costly during inference. Studies also find prompting LLMs with only scalar feedback is not as effective as using both scalar and verbal feedback (Peng et al., 2023). These methods are further limited by unrecognizable language or non-language sequential data sources (for example, DNA, protein, or chemical sequences) due to lack of domain knowledge (Ouyang et al., 2024), motivating the need for general-domain rewriting approach like MACS.

**Iterative Refinement via fine-tuning**   To reduce inference costs, a few studies have demonstrated single attribute improvement across a diverse set of tasks via finetuning approaches for small LMs (Padmakumar et al., 2023; Welleck et al., 2023). Typically, a *corrector*—a small LM—edits the previous response from itself or an external LLM to improve downstream task performance. These correctors are supervised finetuned on edit pairs obtained from off-policy sampling or paraphrasing techniques (mask and infill). We built upon these works to provide a unified framework for fine-grained control of multiple external attributes while only using offline data.

**Data-driven approaches for Protein Engineering**   Designing proteins with desirable functionalities using limited data has been a longstanding challenge in biotechnology. Recent works have successfully leveraged machine learning and deep learning methods on assay-labeled data to find new protein sequences with enhanced properties such as fluorescence, binding affinity, stability, assembling, and net charge content (Hsu et al., 2022; Sinai et al., 2020; Ren et al., 2022; Padmakumar et al., 2023; Kirjner et al., 2024; Sternke

---

[1]We only focus on offline methods which exclusively use preexisting data, thus avoid comparison with online and off-policy RL methods in our study which typically require some form of expensive LM exploration.

& Karpiak, 2023). However, most of these approaches are limited to unidirectional optimization of only a single attribute that may compromise other physicochemical properties. Fine-grained control of protein sequences can allow simultaneous tuning of multiple properties of interest and provide deeper insights into sequence-structure-function relationships of proteins across these properties. For example, understanding the impact on activity (Huang et al., 1996; Guo et al., 2004), fluorescence (Shaner et al., 2007; Amat & Nifosì, 2013), stability (Rabbani et al., 2023; Schlinkmann et al., 2012; Childers & Daggett, 2017), solubility (Bolognesi et al., 2019), assembly (Garcia Seisdedos et al., 2022; Bryant et al., 2021) and binding affinity (Starr et al., 2020; Whitehead et al., 2012) under different physiological conditions.

## 3 Multi-Attribute Constraint Satisfaction

### 3.1 Problem Definition

We aim to solve multi-attribute constraint satisfaction for any sequential data as a multi-step LM rewriting task. Formally, the language model is the actor in the Markov Decision Process (MDP), that learns to navigate across a multi-attribute space defined by a set of attribute evaluators $C = \{c_1, c_2, ..., c_k\}$ (which can be classifier probability, regressor, embedding similarity, protein attribute predictors, etc). All attribute evaluators convert sequential inputs into a scalar value within a finite range ($c_j(.) \in [v_{j,min}, v_{j,max}]$). Each MDP episode begins with the initial state containing a context $x$ (that can be empty), a starting sequence $y_0$ and its attribute location $C(y_0)$ and a set of threshold window constraints $T = \{t_1, t_2, ..., t_k\}$, where $t_j = (t_{j,start}, t_{j,end})$ is the threshold boundary for attribute $c_j$. The rewriting language model $M$ iteratively edits the previous sequence until it satisfies the given threshold constraints, i.e., $P_M(y_{i+1}|x, y_i, C(y_i), T)$.[2] Here, each edit $y_i \rightarrow y_{i+1}$ is considered an action, with a deterministic transition to the next state. During inference, the goal is to generate a series of consecutive edits starting from $y_0$ to $y_n$, such that $C(y_n) \in T$.

### 3.2 MACS Approach

**Edit Pairs Construction**   Even though the rewriting process is inherently multi-step during inference, we can isolate individual edits and train language model rewrite using offline pairs. For example, given any pair of similar sequences $y_a$ and $y_b$ which have distinct attribute locations $C(y_a)$ and $C(y_b)$, we can construct a training instance by asking the language model to edit $y_a \rightarrow y_b$ and artificially selecting threshold windows $T_{a \rightarrow b}$[3] that encourage $M$ to move from $C(y_a)$ towards $C(y_b)$ (Andrychowicz et al., 2017). We can similarly define another training instance going from $y_b \rightarrow y_a$. Assuming $m$ variations of a particular sequence are available $(y_1, y_2, ...y_m)$, we can construct $P_2^m$ trainable *edit pairs* from them. In §4 and §5 we show how we create edit pairs for languages and proteins respectively.

**Constraint Satisfaction Reward**   We want to encourage the rewriter LM to make edits that move closer to the user-provided multi-attribute threshold boundary. If the initial sequence is already inside the target threshold boundaries, we expect the LM to paraphrase the sequence. Based on these two aspects, for each attribute ($c_j(.) \in [v_{j,min}, v_{j,max}]$) and its corresponding threshold boundary ($t_j = (t_{j,start}, t_{j,end})$), we define its constraint satisfaction reward as the sum of the two components,

$$R(y_n, y_o, c_j(.), t_j) = \underbrace{f(c_j(y_n), t_j)}_{\text{Satisfaction Score}} + \underbrace{f(c_j(y_n), t_j) - f(c_j(y_o), t_j)}_{\text{Change in Satisfaction Score}} \tag{1}$$

Here $y_n$ and $y_o$ represent the new and the old sequence respectively, while $f(.) \in [0, 1]$ is the threshold satisfaction scoring function that shows the deviation of the attribute score from its threshold boundary. We set the satisfaction score as 1 if its attribute location satisfies the threshold and linearly decreases to 0 as it moves towards the extreme ends,

$$f(c_j(y), t_j) = \begin{cases} \frac{(c_j(y) - v_{j,min})}{(t_{j,start} - v_{j,min})} & \text{if } c_j(y) < t_{j,start} \\ 1 & \text{if } t_{j,start} \leq c_j(y) \leq t_{j,end} \\ \frac{(v_{j,max} - c_j(y))}{(v_{j,max} - t_{j,end})} & \text{otherwise} \end{cases} \tag{2}$$

---

[2] $C(y_i)$ represents a vector of attribute scores for an intermediate output $y_i$

[3] Selecting the threshold window that satisfies $C(y_b)$ works best for training the language model editor.

---

**Algorithm 1:** Multi-Attribute Constraint Satisfaction Training pseudo code

---

**Data:** Edit Pairs Offline set $D = \bigcup_{x,y_a,y_b}\{(x, y_a, y_b, T_{a\to b})\}$, Attribute Evaluators $C = \{c_1, c_2, ..., c_k\}$,
Initial rewriting language model $M$, SFT steps $N_1$, wBC steps $N_2$, Learning rates $\alpha_1, \alpha_2$

**1** Obtain attribute values for all sequences $y_i \in D$

**2** $M_1 \leftarrow M$

**3 for** $i \leftarrow 1$ **to** $N_1$ **do**

**4**     Sample edit pair $(x, y_a, y_b, T_{a\to b})$ from $D$ (random or k-NN sampling)

**5**     $\mathcal{L}_{SFT}(M_1) = -\ln P_{M_1}(y_b|x, y_a, C(y_a), T_{a\to b})$

**6**     $M_1 \leftarrow M_1 - \alpha_1 \nabla_{M_1} \mathcal{L}_{SFT}(M_1)$

**end**

**7** $M_2 \leftarrow M_1$

**8 for** $i \leftarrow 1$ **to** $N_2$ **do**

**9**     Sample edit pair $(x, y_a, y_b, T_{a\to b})$ from $D$ (random or k-NN sampling)

**10**    $\mathcal{L}_{wBC}(M_2) = -R(y_b, y_a, C, T_{a\to b}) \times \ln P_{M_2}(y_b|x, y_a, C(y_a), T_{a\to b})$

**11**    $M_2 \leftarrow M_2 - \alpha_2 \nabla_{M_2} \mathcal{L}_{wBC}(M_2)$

**end**

**12** Evaluate $M_1$ and $M_2$ with multi-step inference strategies

---

The total multi-attribute reward is defined as the sum of satisfaction reward for all attributes,

$$R(y_n, y_o, C, T) = \sum_{j}^{k} R(y_n, y_o, c_j(.), t_j) \tag{3}$$

**Learning** Given a collection of edit pairs $D = \bigcup_{x,y_a,y_b}\{(x, y_a, y_b, T_{a\to b})\}$ we obtain an LM editor by employing supervised finetuning, e.g., with the negative log-likelihood loss $\mathcal{L}_{SFT}(M) = -\ln P_M(y_b|x, y_a, C(y_a), T_{a\to b})$. To improve beyond the supervised finetuned model, we experiment fine-tuning it further with the offline reward-weighted behavior cloning objective (Norouzi et al., 2016; Junczys-Dowmunt et al., 2018; Wang et al., 2020; Ghosh et al., 2021; Ramachandran et al., 2022; Yang et al., 2023; Feng et al., 2023; Baheti et al., 2024), that directly multiplies the reward with SFT objective $\mathcal{L}_{wBC}(M) = -R(y_b, y_a, C, T_{a\to b}) \times \ln P_M(y_b|x, y_a, C(y_a), T_{a\to b})$. We provide the pseudo-code of the MACS training process in Algorithm 1.

**Multi-Step Reward Prioritized Inference** Satisfying multiple precise constraints $T$ across diverse attributes $C$ is a challenging task for which the editor language model, $P_M(y_{i+1}|x, y_i, C(y_i), T)$, may not get the correct answer in one try. A trivial solution is to employ a best-of-N inference strategy. To improve beyond best-of-N, previous solutions to single attribute iterative improvement propose using an iterative editing strategy, where the rewriter LM generates a trajectory of edits sequentially ($y_0 \to y_1... \to y_n$) (Padmakumar et al., 2023; Welleck et al., 2023). However, this naive editing strategy doesn't interact with the attribute evaluators and cannot verify if the intermediate edits are moving toward the threshold constraints or not. We instead propose maintaining a priority queue of edits using the generalized reward function we defined in equation 3. In this strategy, the LM generated subsequent edit $y_i \to y_{i+1}$ is only retained if it moves closer to the threshold satisfaction, i.e. $R(y_{i+1}, y_0, C, T) > R(y_i, y_0, C, T)$. We call this strategy *prioritized* inference and compare its performance against best-of-N and naive editing.

In the subsequent sections, we extensively and systematically evaluate MACS and baselines on a new Fine-grained Constraint Satisfaction (FINECS) benchmark, which comprises two fine-grained control tasks: Text Style Transfer §4 and Protein Design §5.

## 4 FineCS - Text Style Transfer

While foundational large language models are capable of solving a variety of general language tasks via prompt engineering (Ouyang et al., 2022; OpenAI et al., 2024), they incur large computation overhead during inference and often underperform in directly incorporating external real-value feedback (Peng et al., 2023).

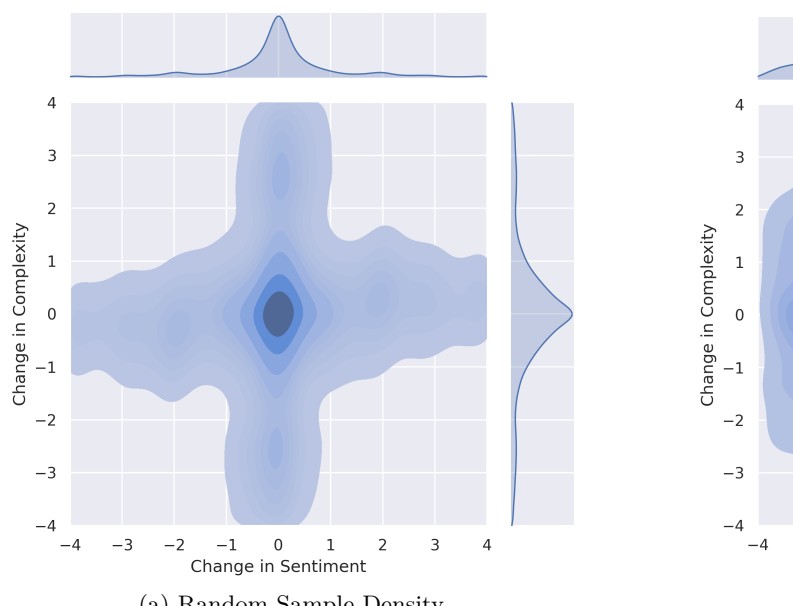
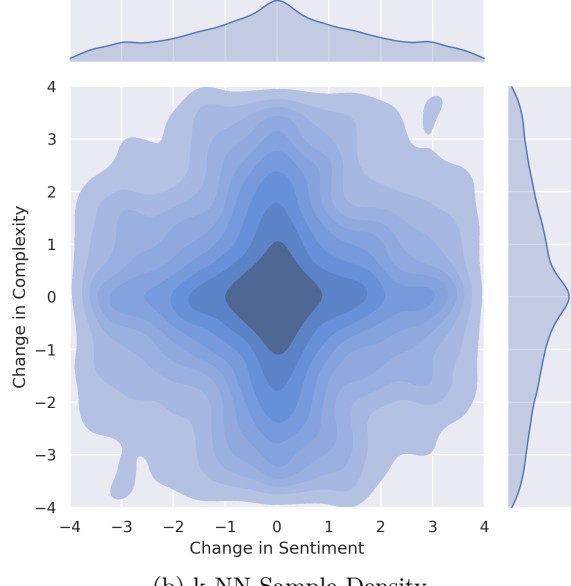

(a) Random Sample Density

(b) k-NN Sample Density

Figure 2: Sentiment and Complexity attribute edit pair distribution via random sampling vs. proposed k-NN sampling. k-NN sampling yields a much more diverse set of edit pairs better suited for simulating editing in all directions.

To mitigate their limitations, we develop MACS to fine-tune small language models that enable constraint satisfaction on external signals via iterative refinement (Padmakumar et al., 2023; Welleck et al., 2023). To evaluate these methods, we create FINECS - Text Style Transfer task, where the goal is to precisely modify the sentiment and complexity of Yelp reviews while preserving fluency and content similarity.

**Attribute Evaluators** To obtain the sentiment and complexity evaluators, we train RoBERTa-large (Liu et al., 2020) regressors on Yelp reviews (Zhang et al., 2015) and the SWiPE Wikipedia simplification dataset (Laban et al., 2023). The output range of the sentiment regressor is $\in [1, 5]$, while the complexity regressor is within the range $\in [-2, 2]$.[4] Subsequently, we defined five threshold boundaries for sentiment as follows: $(1, 1.5)$ very negative, $(1.5, 2.5)$ negative, $(2.5, 3.5)$ neutral, $(3.5, 4.5)$ positive, $(4.5, 5)$ very positive and five threshold boundaries for complexity as follows, $(-2, -1.5)$ very simple, $(-1.5, -0.5)$ simple, $(-0.5, 0.5)$ normal, $(0.5, 1.5)$ complex, $(1.5, 2)$ very complex. In total, these results in 25 different multi-attribute threshold combinations.

We further include two more evaluators to encourage fluency and content preservation: (1) fluency classifier probability $(\in [0, 1])$[5] and cosine text embedding similarity score[6] between the previous and the new output $(\in [0, 1])$. Since we always want to maximize both properties, we add their scores directly in the constraint satisfaction reward function (eqn. 3) as two additional components.

**Creating Attributed Variations and Edit Pairs** To synthetically obtain a diverse set of paraphrases previous studies have proposed various techniques such as mask-then-infill (Xu et al., 2018; Li et al., 2018; Ma et al., 2020; Padmakumar et al., 2023), back-translations (Prabhumoye et al., 2018; Zhang et al., 2018; Lample et al., 2019; Luo et al., 2019), paraphrasing models (Krishna et al., 2020) and generating multiple samples (Welleck et al., 2023). In our preliminary experiments, these methods did not yield many diverse attribute variations. Instead, we use few-shot prompted LLMs to generate alternate attributed variations of reviews. In particular, for both sentiment and complexity attributes, we first sample an equal number of reviews from each threshold boundary ($1K$ from each label, $5K$ total for each attribute). Then, we construct few-shot

---

[4]Training details of sentiment and complexity regressors is provided in Appendix A.1

[5]RoBERTa-base classifier from the Corpus of Linguistic Acceptability (Warstadt et al., 2019) textattack/roberta-base-CoLA

[6]sentence-transformers/all-mpnet-base-v2

LLM prompts that propose five alternate variations of each review, one for each sentiment (or complexity) label. We employ nucleus sampling (Holtzman et al., 2019) on the few-shot prompts ($top_p = 0.95$) to generate 25 variations for each review, $\approx 125K$ total variations for each attribute.[7] Considering the original review and its 25 new proposed variations, we can construct at most $P_2^{26}$ trainable edit pairs for each review (§3.2). The final dataset simply combines all the edit pairs from sentiment and complexity variations. We choose Llama2-7B parameter model (Touvron et al., 2023) as our base LLM and provide the prompts designed for sentiment and complexity attributes in the Appendix A.2.

Within the language domain, edit pairs from synthetic variations are not uniformly distributed. Therefore we propose *k-NN Sampling* to obtain evenly distributed edit pairs in the multi-attribute space. In multi-step editing via LM, consecutive edits can lead to a large drift in content from the original text. Subsequently, we propose an *Anchor Conditioned Inference* strategy to mitigate this problem. We discuss both algorithmic modifications below.

**k-NN Edit Pair Sampling**   An ideal data distribution should have edit pairs from every multi-attribute location to every other location. However, in practice, this is not always true. Given an edit pair $y_a \to y_b$, we visualize its attribute change by converting the difference into vector $C(y_b) - C(y_a)$. We show the distribution of edit pairs when sampled randomly from our synthetic variations in Figure 2a. It shows that most of the edits change only one attribute on average. To obtain a more balanced coverage, we propose using a k-nearest neighbors (k-NN) sampling strategy as follows, (1) sample two multi-attribute threshold boundaries at random, (2) uniformly sample attribute locations from both boundaries (representing start and end) (3) find k-nearest neighbors ($k = 30$) of the sampled transition from the available edit pairs and randomly select one of them. We find the edit pairs sampled with the k-NN strategy to be much more evenly distributed (Figure 2b).

**⚓ Anchor Conditioned Inference**   To reduce content drift in multi-step rewriting, we propose to include the original text in the context of the LM's prompt which we call *anchor conditioning*. During multi-step inference, when a new output sequence is generated, we still retain the original text in the context of subsequent rewrites. To train the rewriter LM with anchor, we augment $D$'s edit pairs ($y_a \to y_b$) by sampling an anchor $y_c$ such that $R(y_c, y_a, C, T_{a\to b}) \geq R(y_b, y_a, C, T_{a\to b})$. In the experiments, we evaluate the effectiveness of this anchor conditioning in content preservation over multiple rewrites.

### 4.1   Text Style Transfer Evaluation

For the Text Style Transfer task on Yelp Reviews, we design a fixed inference budget evaluation setup, i.e., each method will have a fixed number of allowed rewrites to satisfy all multi-attribute constraints. Subsequently, we construct a test set of 250 total reviews (10 from each of the 25 sentiment and complexity threshold combinations). The task is to generate 25 attributed paraphrases for every test review within 5 rewrites ($250 \times 25 \times 5 \approx 31.2K$ total inference budget). For every baseline and our models, we compare the constraint satisfaction success rate of multi-step inference strategies: best-of-N, naive rewriting, and reward-prioritized rewriting. We report the average satisfaction rate, fluency, and embedding similarity of paraphrases that satisfied the given constraints.

**Baselines and MACS models**   As a baseline, we use few-shot prompted Llama2-7B (Touvron et al., 2023) and Llama3-8B (AI@Meta, 2024) models as fine-grained editors for our Text Style Transfer task. For every transition from one threshold combination to another, we find 10 edit pairs as few-shot demonstrations (a total of $25 \times 25 \times 10 = 6250$ edit pairs). For fine-tuning methods, we use a smaller TinyLlama (Zhang et al., 2024) 1.1B parameter model as the multi-attribute rewriter LM. Among finetuning baselines, we compare with Control Tokens (Keskar et al., 2019) that simply convert each threshold combination into style tokens. We allocate 10 total style tokens (5 for each attribute) and simply append the style tokens of the target threshold windows in the prompt along with the target response as follows: $y_a$ [Sentiment

---

[7]The LLM-generated variations do not always agree with the target thresholds but provide a good spread of paraphrases in the multi-attribute space. We filter out variations that yield fluency score or embedding similarity score $< 0.7$.

Table 1: FINECS Text Style Transfer task evaluation: Paraphrase each test review 25 times into fine-grained Sentiment and Complexity threshold constraints while maintaining fluency and content preservation. We compare the Control Tokens baseline with supervised fine-tuned and wBC models each with 3 different inference strategies: Best-of-N, naive rewriting, and reward-prioritized rewriting. We report the average satisfaction rate for each model and inference strategy and average fluency and embedding similarity of the paraphrases that satisfied the constraints. **Takeaway:** Our proposed reward-prioritized rewriting combined with anchor conditioning (⚓) and wBC obtains the highest satisfaction rate. However, its differences with (⚓) and SFT are not statistically significant[†], indicating that anchor conditioning leads to most improvement in multi-step editing.

| | Inference Type | | Best-of-N | | | Naive Rewriting | | | Reward-Prioritized | | |
|---|---|---|---|---|---|---|---|---|---|---|---|
| | Method | Train Sample | Satisfaction Rate* | Flue-ncy | Emb. Sim. | Satisfaction Rate* | Flue-ncy | Emb. Sim. | Satisfaction Rate* | Flue-ncy | Emb. Sim. |
| baselines | 10-shot Llama2-7B | | $.478 \pm .141$ | .93 | .85 | - | - | - | - | - | - |
| | 10-shot Llama3-8B | | $.594 \pm .130$ | .93 | .85 | - | - | - | - | - | - |
| | Control Tokens | random | $.774 \pm .083$ | .92 | .80 | $.783 \pm .068$ | .92 | .78 | $.792 \pm .061$ | .93 | .79 |
| | Control Tokens | k-NN | $.828 \pm .063$ | .92 | .80 | $.809 \pm .046$ | .91 | .78 | $.828 \pm .048$ | .92 | .79 |
| MACS | SFT | k-NN | $.824 \pm .061$ | .92 | .80 | $.809 \pm .056$ | .92 | .78 | $.827 \pm .054$ | .93 | .79 |
| | SFT + wBC | k-NN | $.820 \pm .072$ | .92 | .81 | $.815 \pm .063$ | .92 | .79 | $.835 \pm .051$ | .93 | .80 |
| | ⚓ + SFT | k-NN | $.833 \pm .065$ | .92 | .81 | $\mathbf{.849 \pm .054}$[†] | .92 | .80 | $.847 \pm .052$[†] | .92 | .80 |
| | ⚓ + SFT + wBC | k-NN | $\mathbf{.835 \pm .065}$ | .92 | .81 | $.840 \pm .061$ | .92 | .80 | $\mathbf{.855 \pm .059}$[†] | .92 | .80 |

`Token][Complexity Token]`$y_b$. For LMs trained with MACS, we construct a *text-only* prompt that doesn't use any special tokens as follows:

`Review:` $y_a$
`Review's Sentiment:` $c_1(y_a)$ `and Complexity:` $c_2(y_a)$

`Paraphrase the review such that its Sentiment is within:` $t_{1,a \rightarrow b}$ `and Complexity is within:` $t_{2,a \rightarrow b}$
`Paraphrased Review:` $y_b$

We prepend the above text prompt with $y_c$ and its attribute locations for anchor conditioning (⚓) training.

We train both control tokens and text-prompted models with supervised fine-tuning (SFT) for 200K steps and a batch size of 16. For control tokens, we experiment with both randomized and k-NN edit pair sampling, whereas we only use k-NN edit pair sampling for text-based models. For weighted behavior cloning (wBC) objective, we continue training the supervised finetuned models for an additional 50% steps (100K steps).

## 4.2 Text Style Transfer Results

We present the performance of all baselines and MACS models with the three inference types in Table 1. Among few-shot methods, the newer Llama3 model outperforms Llama2, however, both struggle to achieve very high satisfaction rates and show high variance across different threshold combinations. In comparison, the control tokens-based finetuning baseline works much better than few-shot prompting and gains a further boost in overall satisfaction rate when trained with our proposed k-NN edit pair sampling. Interestingly, naive rewriting is occasionally worse than best-of-N inference, indicating that models may not consistently move toward the threshold boundaries. The reward-prioritized rewriting improves over naive rewriting by leveraging the external scorers and our reward function to guide its search process.

Among our methods, the *text-only* finetuned model matches the performance of the control tokens baseline when trained with the proposed k-NN edit pair sampling. For models without anchor conditioning, we notice that multi-step rewriting can drift away from the original content indicated by a drop in embedding similarity when switching from best-of-N to rewriting. Anchor conditioning (⚓) resolves the content drift problem and subsequently improves satisfaction rate and final embedding similarity when employing rewriting inference strategies. Finally, we notice that models trained with wBC outperform their counterpart SFT-only models

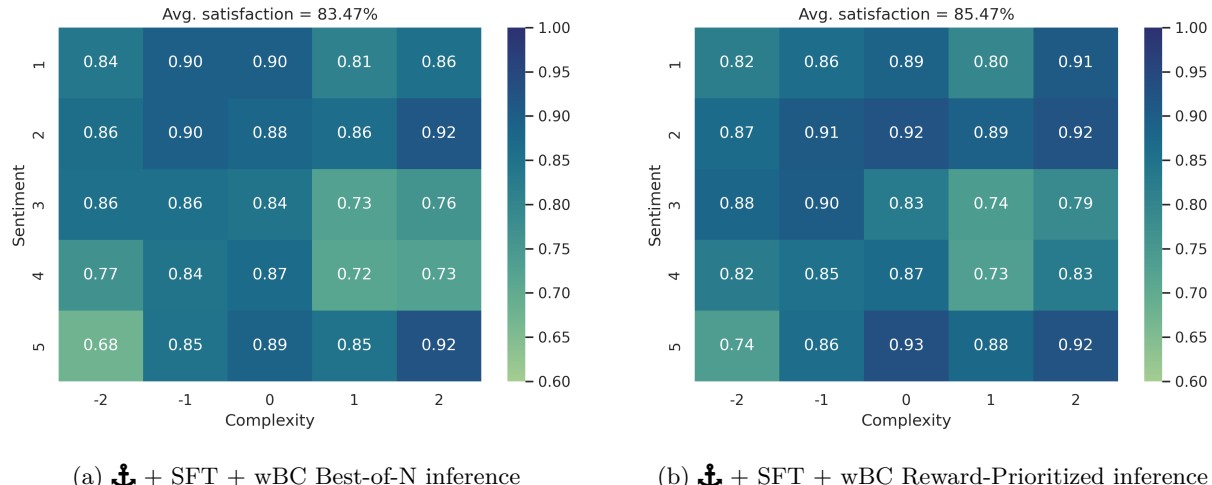

(a) ⚓ + SFT + wBC Best-of-N inference

(b) ⚓ + SFT + wBC Reward-Prioritized inference

Figure 3: Comparing best-of-N vs. reward-prioritized inference constraint satisfaction rate of Sentiment and Complexity attributes. **Takeaway:** Reward-prioritized inference has better satisfaction rates in *harder to reach* constraints i.e. edges of the satisfaction matrix.

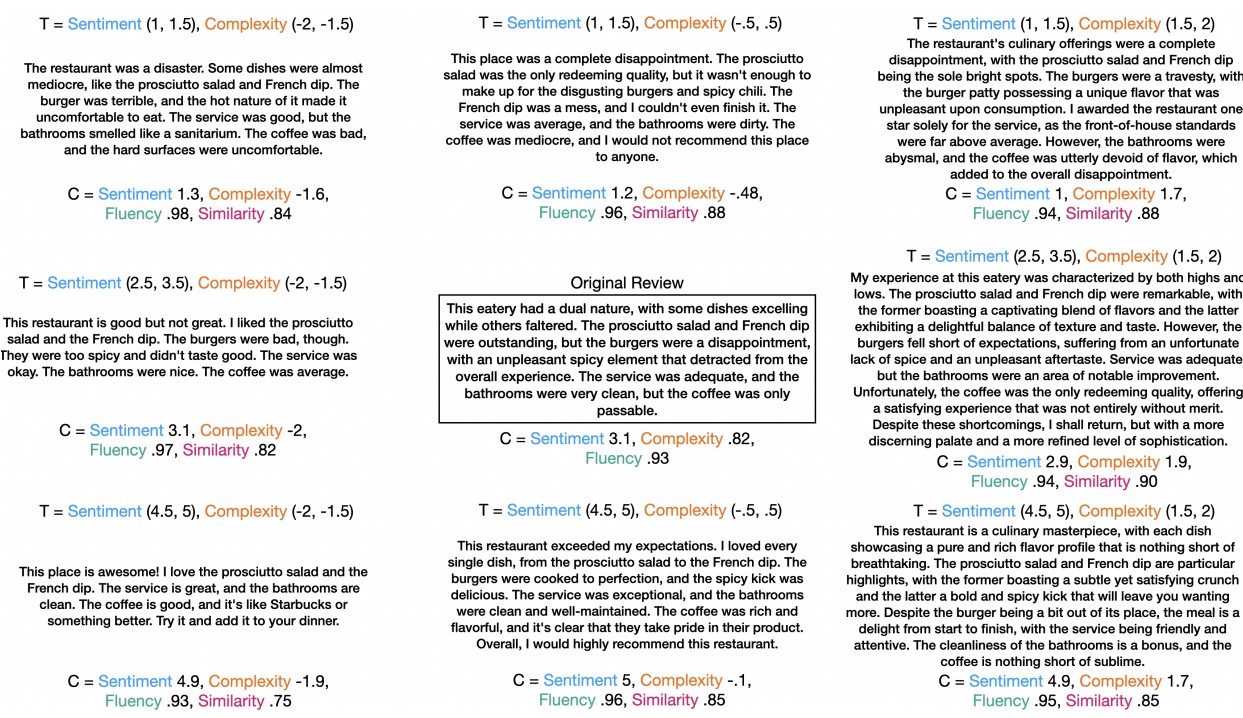

Figure 4: Showing 8 attributed paraphrases of a test review for various thresholds generated by ⚓ + SFT + wBC model with reward-prioritized rewriting.

in reward-prioritized rewriting. However, the performance differences are not statistically significant ($p \approx 0.2$) according to the two-proportions z-test (Fleiss et al., 2013). We show the detailed threshold satisfaction matrix of best-of-N vs. reward-prioritized inference for our wBC model in Figure 3. Reward-prioritized inference achieved a better satisfaction rate than best-of-N in most of the threshold constraints, especially for harder-to-reach threshold constraints (corners of the threshold satisfaction matrix). We also present example generations from the best method in Figure 4, showcasing the difficulty of the task.

# 5  FineCS - Protein Design

Unlike language, where text can be paraphrased in many different ways, fine-grained editing in protein space is challenging due to (a) the uneven distribution of assay-labeled data across multiple attributes and (b) the existence of limited potential solutions in nature for a given set of attribute constraints (Sternke & Karpiak, 2023). Moreover, for any given set of constraints, obtaining multiple novel and diverse candidates is important to maximize the chances of success in wet lab experiments (Jain et al., 2022). To evaluate fine-grained control of MACS framework in protein space, we create FineCS - Protein Design, where the task is to simultaneously modulate fluorescence and folding stability of Green Fluorescent Protein (GFP), (a protein widely investigated and used as biosensors in life sciences research).

**Fluorescence and folding Stability Evaluators**   We obtain the dataset of $\approx 51.7K$ mutants of the GFP *wild-type* (i.e., the protein sequence that occurs in nature) (Sarkisyan et al., 2016; Gonzalez Somermeyer et al., 2022). The dataset contains fluorescence levels on logarithm 10 scale for every mutant sequence $\in [1.28, 4.12]$. Due to a lack of assay-labeled data for a second attribute, we calculate the theoretical folding stability values ($\Delta\Delta G$ or ddG) of each mutant with respect to the wild-type structure using FoldX software(Schymkowitz et al., 2005).[8] The wild-type ddG is 0 and any mutant with negative ddG is more stable than wild-type. The overall distribution of ddG for all the mutants is $\in [-5.66, 60.75]$. We train ESM2-based regressors (Lin et al., 2023) as evaluators for both attributes using mean squared error loss.[9] The test set correlation for fluorescence and ddG are 0.974 and 0.987 respectively.[10]

**Attribute Distribution and Edit Pairs**   We plot the distribution of log fluorescence and ddG of all the GFP mutations in Figure 8 in the Appendix. Unlike language data, protein mutants are even more unevenly distributed across the multi-attribute landscape, with the bulk of the mutants clustered near the wild-type (WT) GFP sequence (which has $\approx 3.72$ log fluorescence and 0 ddG). To effectively navigate this skewed distribution, we define four threshold boundaries in log fluorescence, $(< 3.0)$ - very low, $(3.0, 3.4)$ - low, $(3.4, 3.7)$ - medium, $(> 3.7)$ - bright and four threshold boundaries in ddG, $(< 0)$ - more stable than WT, $(0.0, 0.5)$ - as stable as WT, $(0.5, 2.0)$ - slightly destabilized, $(> 2.0)$ - highly destabilized (Dill et al., 2008).

The limited viable solutions in certain regions ($< 10\%$ of proteins have $< 0$ ddG) make the protein design task very challenging, especially when learning from an offline dataset of mutations. Here, all GFP mutants are considered *paraphrases* of each other, and thus, total possible edit pairs are $\approx P_2^{51.7K}$. To train the LM rewriting models to edit in all possible directions, we employ the following edit pair sampling strategy: (1) pick two multi-attribute threshold boundaries, (2) sample a mutant at random from both of the selected threshold constraints and (3) construct an edit pair by treating the first as the source and the second as the target mutant.

## 5.1  Protein Design Evaluation

Unlike the Style Transfer task, where we only care about one solution for each constraint, the goal of the Protein Design task is to find the maximum number of new mutants in every multi-attribute constraint under a fixed inference budget. For each threshold constraint, we initiate multiple random walks of different lengths starting from wild-type GFP sequence ($WT \rightarrow y_1... \rightarrow y_n$). We assign a total 3000 inference budget which results in (1) $3000 \times$ 1-hops, (2) $1000 \times$ 3-hops, and (3) $300 \times$ 10-hops random walks. We expect duplicated predictions under specific constraints since certain regions will have naturally very few solutions. Among the 3000 predictions in each inference method, we calculate the *total success rate*: ratio of distinct mutants that satisfy the constraints according to our evaluators and *unique success rate*: ratio of unique successful mutants outside of the offline training data. We also compare with reward-prioritized walks from wild-type (§3) where the LM generated intermediate edit $y_i \rightarrow y_{i+1}$ is only retained if it moves closer to the threshold constraints, i.e. $R(y_{i+1}, WT, C, T) > R(y_i, WT, C, T)$. We experiment with reward-prioritized walks in $1000 \times$ 3-hops

---

[8]Foldx uses an empirical force field to determine the effect of mutations on the protein folding. We note that FoldX-generated values are not an accurate representation of real experimental folding stability and are only used as a proxy. We follow best practices recommended in the previous research and compute ddG on an average of five FoldX calculations (Chan et al., 2021).

[9]We divide the dataset into 50% train, 15% validation, and 35% test set for both fluorescence and ddG attributes.

[10]Implementation details in Appendix B.1

Table 2: FINECS Protein Design task evaluation: Starting from GFP wild-type, discover the maximum possible unique mutants across 16 multi-attribute constraints of log fluorescence and ddG within 3000 total inferences. We compare the ProtGPT2 LM editor fine-tuned with SFT and wBC with 5 different inference strategies: three random walks and two reward-prioritized walks with different hop lengths. After discarding all duplicate solutions, we report the average rate of mutants that satisfy the threshold constraints (total success rate) and the average rate of successful mutants that are outside the training data (unique success rate). We also report the average edit distances between all pairs of successful mutants for each method. *Takeaway:* wBC \w entropy, another variant of MACS method, discovers the most number of novel mutants. However, the differences between different inference methods are not statistically significant[†].

| | Total Success Rate | Unique Success Rate* | Edit Distance | Total Success Rate | Unique Success Rate* | Edit Distance | Total Success Rate | Unique Success Rate* | Edit Distance |
|---|---|---|---|---|---|---|---|---|---|
| | | Random | | | Recombine | | | Unique Recombine | |
| baseline | 8.3 | 8.2 | $4.4 \pm 2.2$ | 36.5 | 30.0 | $3.9 \pm 1.6$ | 39.5 | 39.5 | $3.8 \pm 1.6$ |
| | | SFT | | | SFT + wBC | | | SFT + wBC \w entropy | |
| random walk 3000 × 1-hop | 41.3 | 38.6 | $4.2 \pm 2.0$ | 43.6 | 40.2 | $4.0 \pm 1.9$ | 44.3 | $41.1^{†}$ | $4.4 \pm 2.3$ |
| random walk 1000 × 3-hop | 41.3 | 38.6 | $4.2 \pm 2.1$ | 43.6 | 40.1 | $4.0 \pm 1.9$ | 44.5 | $41.2^{†}$ | $4.5 \pm 2.3$ |
| random walk 300 × 10-hop | 41.8 | 39.0 | $4.2 \pm 2.1$ | 43.8 | 40.4 | $4.0 \pm 1.9$ | 44.7 | $\mathbf{41.5^{†}}$ | $4.5 \pm 2.4$ |
| priority walk 1000 × 3-hop | 41.6 | 38.9 | $4.2 \pm 2.0$ | 43.5 | 40.1 | $4.0 \pm 1.9$ | 44.4 | $41.1^{†}$ | $4.5 \pm 2.3$ |
| priority walk 300 × 10-hop | 41.1 | 38.4 | $4.2 \pm 2.1$ | 43.5 | 40.1 | $4.1 \pm 1.9$ | 44.6 | $\mathbf{41.5^{†}}$ | $4.4 \pm 2.3$ |

and 300 × 10-hops settings. In total, for 16 multi-attribute threshold constraints of log fluorescence and ddG, we have a total budget of $16 \times 3K = 48K$ decoding in every inference method.

**Baselines** We compare our method with two baselines: (1) Random - proteins are randomly mutated based on the edit-distance distribution of the train-set sequences and (2) Recombine - a previous method that samples new diverse sequences by shuffling and merging pairs from an initial seed set (Otwinowski et al., 2020; Sinai et al., 2020). Recombine can sample many duplicate sequences when the seed set is small (certain threshold combinations have fewer than 100 original sequences). We also compare with a stronger variant of Recombine where we ensure that every 3000 newly sampled sequences for a specific multi-attribute constraint are unique and outside the seed set. We call this stronger baseline Unique Recombine.

**MACS training** We finetune the ProtGPT2 LM (Ferruz et al., 2022), which is a 738M parameter protein language model,[11] as the rewriter for this task. Since ProtGPT2 does not have English words as tokens, we prompt the mutant edit pair to the LM as follows: $y_a$[A Fluorescence]$c_1(y_a)$[A ddG]$c_2(y_a)$[Target Fluorescence]$t_{1,a \to b}$[Target ddG]$t_{2,a \to b}$[edit]$y_b$, where intermediate key-words are special tokens added to the model's vocabulary. Using the edit pair sampling strategy described earlier, we train ProtGPT2 with SFT for 20K steps with batch size 16 and learning rate $10^{-4}$. We then further continue finetuning with the wBC objective for an additional 10K steps and learning rate $10^{-5}$. Since we want to encourage the LM editor to generate diverse candidates in this task, we separately also train with wBC objective augmented with entropy penalty (coefficient $\gamma = 0.05$).[12] We present additional implementation details for our methods and the baselines in Appendix B.2

## 5.2 Protein Design Results

We report the evaluation results of baselines and different variants of MACS in Table 2. Random mutation shows the worst performance as expected, while Recombine is a strong baseline that finds more unique and successful mutants. With Unique Recombine, we establish an upper bound on the baseline's performance by only retaining unique sequences. However, the offline wBC model outperforms both the Recombine baseline and the SFT model across all inference strategies. When augmented with entropy penalty, we observe a boost in success rate for the wBC model and a larger spread of edit distances indicating more diverse

---

[11]https://huggingface.co/nferruz/ProtGPT2
[12]$\mathcal{L}_{wBC \ \w \ entropy}(M) = \mathcal{L}_{wBC}(M) + \gamma(P_M(y_b|x, y_a, C(y_a), T_{a \to b}) \ln P_M(y_b|x, y_a, C(y_a), T_{a \to b}))$

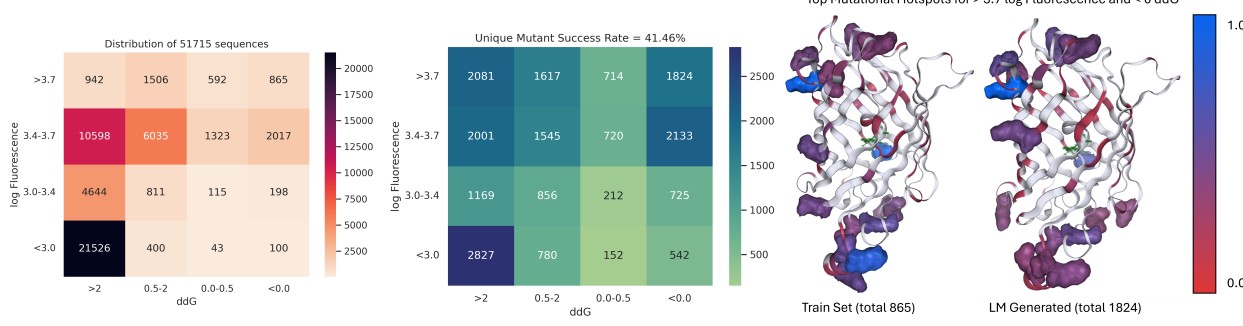

(a) Train Mutant Distribution

(b) SFT + wBC \w entropy - priority 300 × 10-hop

(c) Comparing Mutational Hotspots

Figure 5: Analyzing new mutant discovery of reward-prioritized walk compared with training set distribution. **Takeaway:** Even with very few training instances in many of the regions, LMs trained with MACS discover many novel candidates. Analysis of the top 15 mutational hotspots reveals that LMs can extrapolate beyond the mutational patterns seen in the training set.

mutants. Although reward-prioritized and naive multi-hop walks yield the best performance, we do not notice a significant difference between different inference strategies with two proportions z-test.

Finally, we compare the distribution of train set mutant vs. the newly discovered mutants via the reward-prioritized walk (300 × 10-hop) from wBC \w entropy model in Figure 5. Despite small sample sizes of training data in certain regions, our method can extrapolate beyond the original training set and find diverse sequences even with offline training. Finally, when comparing mutational hotspots and their distribution across GFP structure, our LM-generated sequences show a different distribution and occasionally novel mutations compared to train set sequences, as shown in Figure 5c (and Figure 9 in the Appendix).

# 6 Conclusion

We create Multi-Attribute Constraint Satisfaction (MACS) framework to cheaply train LMs as fine-grained editors by sampling edit pairs from offline sequential datasets. We also create a new Fine-grained Constraint Satisfaction (FINECS) benchmark to evaluate our method, comprising two challenging fine-grained controllability tasks. In the FINECS Text Style Transfer task, LM editors trained with weighted behavior cloning paired with proposed k-NN edit pair sampling, and multi-step reward-prioritized editing outperform their SFT counterparts and other inference methods. We boost its performance further with anchor conditioning and achieve the highest constraint satisfaction rates compared to previous fine-tuning and few-shot prompted baselines. Interestingly, in the FINECS Protein Design task, MACS can train protein language models to discover novel proteins outside the training data with high success rates while highlighting different mutational hotspots. Our study demonstrates the potential of LMs as fine-grained writing assistants and protein engineering models that can aid in the creation of novel proteins with fine-grained properties.

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

# A FineCS - Text Style Transfer Implementation Details

## A.1 Sentiment and Complexity Regressor Training

**Sentiment Regressor** We train Sentiment regressor on Yelp reviews (Zhang et al., 2015). The original data contained $650K$ train and $50K$ test reviews divided evenly across five labels (1 - very negative, 2 - negative, 3 - neutral, 4 - positive, and 5 - very positive). We filter reviews that are non-English[13] or long ($> 200$ tokens). After filtering, we obtain $\approx 464K$ train reviews and $\approx 36K$ test reviews. We randomly sample $\approx 36K$ reviews from the train set for validation and train a RoBERTa-large (Liu et al., 2020) regressor on the remaining instances for 4 epochs using mean squared error loss. The final regressor obtained a 0.92 test correlation (and 0.37 mean absolute error). During inference, we clamp the predictions from the regressor such that its output range is $\in [1, 5]$.

**Complexity Regressor** To obtain the Complexity regressor we train a ranking model on top of the SWiPE Wikipedia simplification dataset (Laban et al., 2023). The SWiPE dataset contains $\approx 143K$ pairs of simple to complex Wikipedia paragraphs. However, many instances were low quality (very long, very short, high repetition, bad words, non-English, etc.). After filtering these instances, we are left with $79K$ train, $1K$ validation, and $1.8K$ test simple to complex pairs. We train a RoBERTa-large (Liu et al., 2020) regressor on this pairwise data using the ranking objective (Collins & Koo, 2005) for 8 epochs. The best checkpoint emitted raw scores in the range of $\in [-17.1, 17.1]$ and obtained 98.2% accuracy on the test set (comparing the raw scores of simple and complex passage pairs). We linearly interpolate this output range to be $\in [-2, 2]$ such that we can subdivide the output range from the regressor into five fine-grained threshold boundaries (to match the Sentiment regressor labels).

---

[13]Using external language identification classifier `https://huggingface.co/papluca/xlm-roberta-base-language-detection`.

```
[INST] <<SYS>>

You are an advanced stylistic paraphrasing AI that is designed to generate high-
quality, grammatically correct, non-repetitive, semantically similar and
stylistically diverse paraphrases. Follow the user's prompt structure precisely and
retain most of the lexical and topic content of the original input when changing the
style.

<</SYS>>

You are a Sentiment changing paraphraser for yelp reviews. When paraphrasing, do not
deviate too far from the original review in terms of lexical and topic coverage.
Reviews on yelp contain 5 levels namely: 1 - (strongly negative), 2 - (negative), 3 -
(neutral), 4 - (positive), 5 - (strongly positive). Given a human written review of a
particular level, modify it to generate 5 variations for each sentiment level (one
per line) as follows:

Original Review: <review>

Sentiment Level: <level>

Variation 1 (strongly negative): <variation1>

Variation 2 (negative): <variation2>

Variation 3 (neutral): <variation3>

Variation 4 (positive): <variation4>

Variation 5 (strongly positive): <variation5>

[/INST]
```

Figure 6: Llama2 Sentiment paraphrasing prompt

## A.2  Sentiment and Complexity Few-Shot Prompts

To generate attributed variations of Yelp reviews in the Sentiment and Complexity axis we use few-shot prompting on top of a Llama2-7B[14] parameter model (Touvron et al., 2023). The prompt used for Sentiment and Complexity are given in Figures 6 and 7 respectively. We augment both prompts with their own 3-shot demonstrations and generate 5 samples for each review using nucleus sampling ($top_p = 0.95$).

## B  FineCS - Protein Design Implementation Details

### B.1  Fluorescence and ddG Regressor Training

To train the protein evaluators, we randomly split the $\approx 51.7K$ mutant sequences into 50% train, 15% validation, and 35% test sequences. Along with the ESM2 model-based regressors, we also experimented with traditional CNN regressors (Dallago et al., 2021). The learning rate for both models is 1e-4 where the ESM2-based regressor is trained for 12 epochs and the CNN regressor was trained for 40 epochs. Despite additional training time, the test set correlation for fluorescence and ddG for the CNN regressors are 0.892 and

---

[14]meta-llama/Llama-2-7b-chat-hf

```
[INST] <<SYS>>

You are an advanced stylistic paraphrasing AI that is designed to generate high-
quality, grammatically correct, non-repetitive, semantically similar and
stylistically diverse paraphrases. Follow the user's prompt structure precisely and
retain most of the lexical and topic content of the original input when changing the
style.

<</SYS>>

You are a Linguistic Complexity changing paraphraser for yelp reviews. When
paraphrasing, do not deviate too far from the original review in terms of lexical and
topic coverage. Reviews on yelp can vary on 5 levels of complexity namely: 1 - (very
simple), 2 - (simple), 3 - (normal), 4 - (complex), 5 - (very complex). Given a human
written review of a particular level, modify it to generate 5 variations for each
complexity level (one per line) as follows:

Original Review: <review>

Complexity Level: <level>

Variation 1 (very simple): <variation 1>

Variation 2 (simple): <variation 2>

Variation 3 (normal): <variation 3>

Variation 4 (complex): <variation 4>

Variation 5 (very complex): <variation 5>

[/INST]
```

Figure 7: Llama2 Complexity paraphrasing prompt

0.933 respectively, that are much lower than ESM2-based models (0.974 and 0.987 respectively). Subsequently, we use the ESM2-based regressors as evaluators in our experiments.

## B.2 Protein Design Baselines and LM editor Implementation

The $51.7K$ GFP train mutants are unevenly divided across the 16 multi-attribute threshold combinations as seen in Figure 5a. In the Random mutation baseline, when predicting new mutant sequences from a particular threshold combination, we maintain the edit distance distribution of the train sequences within the same threshold combination. The Recombine baseline uses a recombination strategy where a pair of sequences are mixed (shuffling each position with a recombination rate $\kappa = 0.5$) to create two new sequences. When generating new sequences for a particular threshold combination with the Recombine baseline, we set the train sequences within the same thresholds as the seed set, randomly shuffle them, and iteratively apply the recombination strategy until we get 3000 new sequences. Since some threshold combinations have very low seed sequences ($<200$) there may be duplicates when generating the 3000 new sequences with this strategy. We improve upon this baseline, we create Unique Recombine where we keep generating sequences with the recombination strategy until we get 3000 unique sequences that don't overlap with the training set.

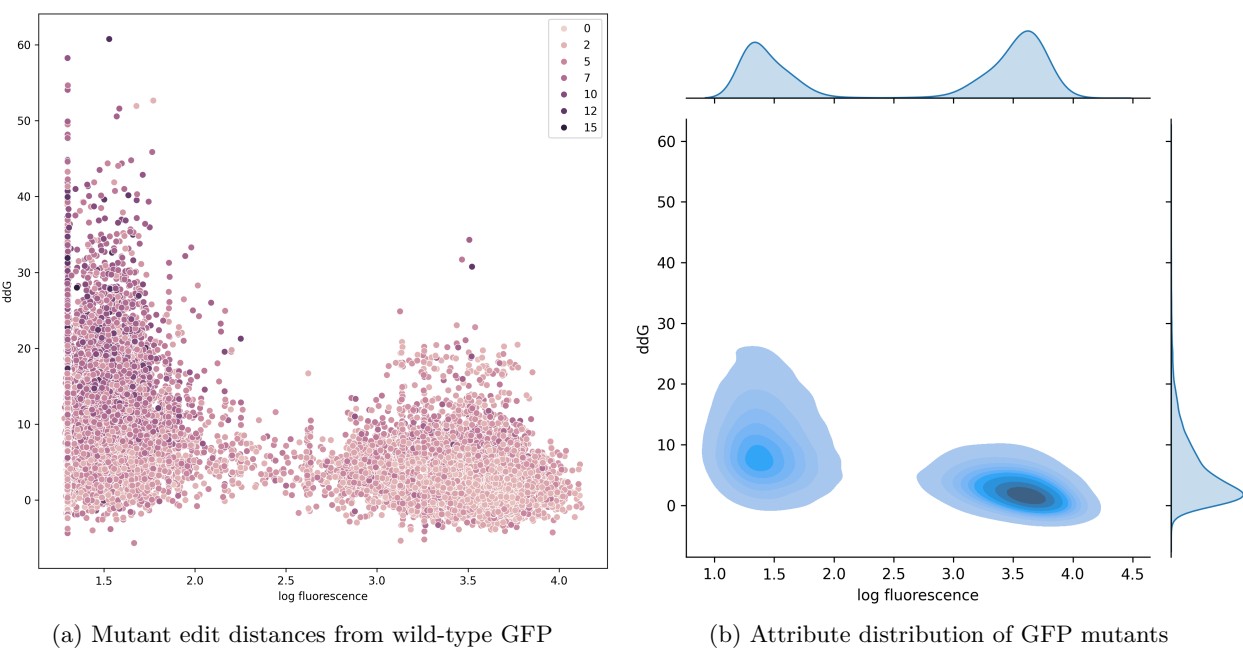

(a) Mutant edit distances from wild-type GFP

(b) Attribute distribution of GFP mutants

Figure 8: Log Fluorescence and ddG distribution of $51.6K$ GFP mutants

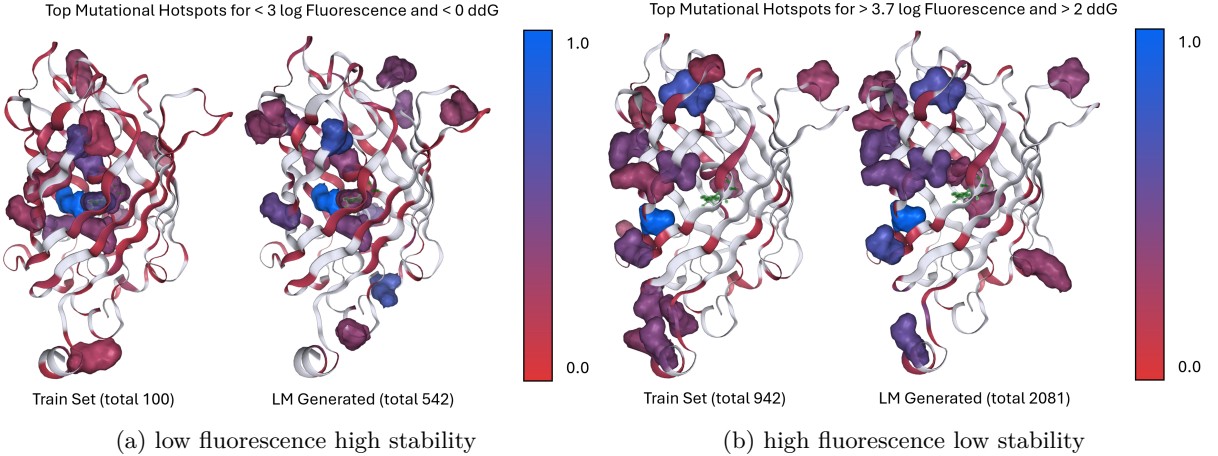

(a) low fluorescence high stability

(b) high fluorescence low stability

Figure 9: Comparing the top 15 mutational hotspots from the train set vs. LM predicted demonstrating that MACS can extrapolate beyond the mutational patterns seen during training.

For the Protein Language Model editor models trained with our MACS framework, we use the nucleus sampling (Holtzman et al., 2019) with ($top_p = 0.95$). We also had to increase the generation temperature to 1.2 to encourage more diverse sequences. During our early experiments, we sampled edit pairs from each threshold combination uniformly, leading to overfitting in the low-density regions of multi-attribute space, i.e., most LM-generated sequences are duplicated. To mitigate this behavior, we downsampled the threshold combinations containing fewer than $\tau = 400$ sequences.[15]

---

[15]If a target threshold combination is $n < \tau$ sequences, we reduce its edit-pair sampling weights to $n/\tau$ to reduce overfitting in the sparse region.

## C   Limitations and Future Work

MACS is an easy-to-implement framework to train domain-specific language models as fine-grained editors in an offline setting. However, there are a few limitations. Due to the offline nature of our method and our sampling strategy, it is unable to extrapolate well to regions within the multi-attribute space with low or no data points. To train good multi-attribute LM editors, MACS requires a good initial domain-specific pretrained language model. In our preliminary experiments with antibody generation task (Wong et al., 2023), a chemistry LM[16] trained with MACS was not able to generate many novel candidates, likely due to its small size and poor data coverage.

In the future, we aim to extend our method such that it can use both offline and on-policy samples to improve its performance and diversity in the fine-grained control task. Further research is also needed to support categorical and lexical constraints in MACS.

---

[16]https://huggingface.co/ncfrey/ChemGPT-19M

