# OpenReview forum: "Multi-Attribute Constraint Satisfaction via Language Model Rewriting"
_TMLR — Accepted by TMLR_

### Review · Reviewer_JCoR · 2024-08-16

**Summary Of Contributions:**

The paper introduces a framework, Multi-Attribute Constraint Satisfaction (MACS), which aims to enhance the capability of language language models (LLMs) to satisfy multiple user-specified constraints on external real-value attributes. The authors propose a method that involves fine-tuning LLMs in any sequential domain, leveraging weighted behavior cloning to iteratively improve outputs and meet constraints across various attributes. The experimental results on text style transfer and protein design tasks demonstrate MACS's effectiveness in achieving high constraint satisfaction rates, outperforming existing domain-specific baselines.

**Audience:**

Yes

**Claims And Evidence:**

Yes

**Requested Changes:**

1. Hard Constraints Satisfaction: Can the MACS method strictly satisfy hard constraints, or are there scenarios where it may fail to meet these constraints? If so, how does the method handle such failures?

2. Offline RL Formulation Clarification: The paper mentions using Offline Reinforcement Learning for training the language model editor. Could the authors clarify how the rewriting process is formulated as a Markov Decision Process? It seems that the method described leans more towards imitation learning via Weighted Behavior Cloning, which is traditionally distinct from offline RL approaches [1]. How does this method reconcile with the principles of offline RL?

3. Comparison with Self-Prompting: The paper could benefit from a comparison with multi-round self-prompting, which are also designed to iteratively improve language model outputs. How does MACS differ from or compare to such self-prompting methods in terms of effectiveness and efficiency?

4. Emphasis on Real-Value Attributes: The paper appears to place significant emphasis on the satisfaction of real-value attributes constraints. Reflecting this focus in the title or abstract could potentially highlight the paper's contributions more clearly. Would the authors consider adjusting the title to better reflect this unique aspect of their work?

[1] Rashidinejad, Paria, et al. "Bridging offline reinforcement learning and imitation learning: A tale of pessimism." Advances in Neural Information Processing Systems 34 (2021): 11702-11716.

**Strengths And Weaknesses:**

* Strengths:

1. The paper presents a robust solution for Real-Value Attributes Constraint Satisfaction, a significant challenge in the field. The MACS framework's ability to handle step-by-step refinement for such constraints is particularly noteworthy and addresses a crucial gap in current methodologies.

* Weaknesses:

1. One of the weaknesses is the difficulty in constructing the datasets needed for training the step-by-step refinement process. The paper would benefit from a more detailed discussion on how such datasets can be effectively created and utilized.

2. The paper could improve in detailing the baseline configurations, and algorithmic procedures. Providing a step-by-step pseudo code would enhance understanding and reproducibility of the methods described.

3. While the experiments conducted are thorough within the presented tasks, expanding the scope to include more tasks would strengthen the paper's claims. Demonstrating the framework's versatility across a broader range of applications would be beneficial.

---

> ### Author Response · Authors · 2024-10-27
> **Rebuttal for Reviewer JCoR**
>
> We sincerely appreciate your thoughtful review and insightful comments on our manuscript. We are grateful for your acknowledgment of MACS's effectiveness in achieving high constraint satisfaction rates and outperforming domain-specific baselines. Your recognition of our work's strengths is truly encouraging.
>
> We would like to address your comments and suggestions:
>
> 1. **Difficulty in Dataset Construction**: We recognize the challenge of dataset construction for multi-attribute iterative refinement but have a few promising solutions. For language tasks, we leverage synthetic data curation using 7B-parameter language models, with the potential to scale to larger LLMs like GPT-4 given more resources. In non-language domains such as small molecules and proteins, we can utilize random mutations, molecular dynamics models, and specialized softwares to augment data with unseen sequences and external attributes. To generate diverse synthetic candidates, MACS can further leverage modern state-of-the-art Protein[1] and Chemistry Language Models[2], significantly expanding limited datasets. These strategies can enhance MACS's generalizability and robustness across diverse applications.
>
> 2. **Hard Constraints Satisfaction**: You raise an important point regarding hard constraints. As briefly mentioned in Appendix C, "Limitations and Future Work", MACS's offline nature indeed limits its ability to extrapolate to regions where data is unavailable. For instance, designing proteins that are simultaneously extremely fluorescent and unstable would be challenging due to the lack of such examples in real life and the data. MACS’s inference search would fail to yield any constraint satisfying candidates and we acknowledge this as a limitation of the method. In the future, we aim to extend our approach and leverage both offline and on-policy learning to mitigate this limitation.
>
> 3. **Implementation Details**: We appreciate your suggestion for more detailed implementation information. We have provided comprehensive details of our method and baselines in Appendix B.2. Additionally, following Reviewer VzBx's suggestion, we will include an algorithm box in the revised manuscript to enhance clarity and reproducibility.
>
> 4. **Offline RL and Weighted Behavior Cloning**: We appreciate your astute observation regarding the distinction between offline RL and our approach. While we initially framed the problem within an RL setup to accommodate reward-guided inference, our offline learning method primarily employs supervised fine-tuning and weighted behavior cloning. These techniques incorporate elements of learning from reward-weighted edits but differ from traditional offline RL. We will revise the manuscript to make this distinction explicit and avoid any potential confusion.
>
> 5. **Comparison with Self-Prompting**: Your suggestion to compare MACS with self-prompting methods is valuable. However, it's worth noting that most self-prompting and iterative improvement literature relies on very large language models (e.g., GPT-4) [3], which are even more computationally expensive and time-consuming than our Llama 3 8B baseline. In contrast, MACS utilizes more efficient fine-tuned TinyLlama 1B models, achieving superior performance with significantly reduced computational costs (<1 hr total inference time compared to the Llama 3 model which has more than 10 times inference cost on A40 GPU). Moreover, prompting strategies may not apply to non-language domains such as proteins or DNA, where MACS demonstrates its versatility.
>
> 6. **Emphasis on Real-Value Attributes**: We appreciate your suggestion to highlight our focus on real-value attributes. As mentioned in Appendix C, our future work aims to extend MACS to handle both offline and online data, as well as incorporate additional constraint types (e.g., categorical, lexical). We will update the abstract and introduction to better reflect our current emphasis on real-value attributes.
>
> Once again, we sincerely thank you for your valuable feedback, which will undoubtedly help us improve our manuscript. We look forward to addressing these points in our revised version.
>
> **References**:
> [1] (Nguyen et. al 2024) Sequence modeling and design from molecular to genome-scale with Evo https://www.biorxiv.org/content/10.1101/2024.02.27.582234v1
> [2] https://huggingface.co/yerevann/chemma-2b
> [3] (Madaan et. al, 2023) Self-refine: Iterative refinement with self-feedback https://arxiv.org/abs/2303.17651

---

### Review · Reviewer_VzBx · 2024-08-28

**Summary Of Contributions:**

The paper proposes to carry out rewriting using LLM to solve constraint satisfaction problems for certain text based domains, such as text style transfer and protein designs. The paper proposes a RL approach: they describe a dataset collection process, reward design as well as loss and algorithm for offline RL followed by SFT. They extensively evaluate their approach with a number of datasets and show improvements over baselines.

**Audience:**

Yes

**Claims And Evidence:**

No

**Requested Changes:**

=== *Presentation clarity* ===

Overall, I find the paper lacks clarity in Presentation. The description of the offline RL method should be better highlighted in the presentation and draws connection to the weighted BC literature. In its current form, the paper spends lots of text for description rather than formula description, making it a bit harder for technical readers to follow. An algorithm box would help.

=== *Reward design* ===

The reward design process is critical for offline RL. From the description of the work, the authors propose to label randomly chosen pairs of data with some instructions such that they can be understood as some form of text rewriting data. I find this process to be remindful of hindsight experience replay [1] line of work, which can be of interest to discuss.

[1] Andrychowicz et al, Hindsight experience replay, 2017

=== *Overall design* ===

My overall concern with the paper is the technical depth and solidity of the overall method, since using LLM for rewriting itself is not a very novel application, and despite the interesting design on data collection and reward design, it feels more like a bag-of-tricks rather than a coherent framework to solve the constraint satisfaction problem. The paper also discusses lots of specific points regarding applications, such as domain-specific designs for protein design and text style transfer, e.g., knn-sampling seems to be coupled wiht text style transfer. This makes it a bit difficult to disentangle what constitutes a more general algorithmic design of the paper, and what constitutes an application-specific design choice.

=== *Evaluation for text transfer* ===

In text style transfer in table 1, we find the new approach generally outperforms the previous baseline. However, I notice that SFT itself is already bringing most of the benefits and in fact in certain cases wBC (offline RL) does not add much on top of SFT, or the improvement from SFT to SFT + wBC is generally not statistically significant (within error bar, despite highlighted in the table). This paints the general picture that SFT is the main contributor to the performance gain, which makes sense since all other baselines do not make use of domain-specific SFT fine-tuning. It then poses the question of whether wBC or offline RL is needed at all in this process, since SFT feels like all you need.

I think the evaluation comparison here should highlight the importance of wBC over SFT, because SFT is a common process and performance improvement is expected compared to few-shot or control token baselines. wBC is a key contribution of this paper, whose role is not clear from the current eval.

=== *Evaluation for protein design* ===

In protein design eval in table 2, it is nice that wBC yields general gains over SFT. But it is interesting to see that entropy regularization seems to generally increase the edit distance (contrary to wBC alone). Can you elaborate more on this?

Can you also provide error bar for the evaluation in order to assess statistical significance.

**Strengths And Weaknesses:**

The strength of the paper lies in that it seems that using LLM to do rewriting for constraint satisfaction of multiple attributes in text-based domains seems to be an interesting and relatively under-explored application. Despite the fact, that using LLM for rewriting is itself not a super novel design choice.

The main weakness might be that the technical evaluation does not seem to yield significant gains over certain baselines in the empirical comparison, posing the question as to whether offline RL is needed at all in this process or does SFT suffice. As mentioned above, LLM rewriting is itself not a very novel design choice, and the technical contribution of the paper relies a lot on the algorithmic part of the empirical design, which seems to be a bit lacking in the current paper. The presentation can also use improvement in its clarity.

---

> ### Author Response · Authors · 2024-10-27
> **Rebuttal for Reviewer VzBx**
>
> We appreciate your thorough review and valuable feedback. We'd like to address your concerns and clarify some aspects of our work. Due to the limited text capacity, we divide the full rebuttal into two comments.
>
> 1. **Main Contribution**: We thank the reviewer for highlighting that our work addresses an _"interesting and relatively under-explored application"_. To our knowledge, we are the first to successfully apply a carefully designed framework to this complex real-world task using only offline data, with encouraging outcomes. Our primary contribution is demonstrating the MACS method to fine-tune small language models on existing corpora in sequential domains and leverage inference-time computing to satisfy precise fine-grained constraints with high accuracy, a task that even 8B parameter LLMs struggle with. MACS consistently outperforms strong domain-specific baselines in both text and protein tasks. To enable future work in this direction, we create a new Finegrained Constraint Satisfaction benchmark with two challenging editing tasks: Text Style Transfer and Protein Design, each with finely divided multi-attribute constraints.
>
> 2. **Evaluation for Text Style Transfer and Protein Design Tasks**: We sincerely appreciate the reviewer's perceptive observations regarding the performance differences across our tasks. To address these concerns, we conducted two-proportion z-tests [1] comparing the overall success rates (across all test instances and thresholds) of the best model and inference strategies with others for both tasks.
>
>    In the text style transfer task, when looking at models without anchor conditioning, wBC with reward-prioritized inference obtains better performance than SFT and naive editing/best-of-N inference strategy. This suggests, that wBC better aligns the editor LM to work with reward-guided inference. However, with anchor conditioning, even though wBC + reward-prioritized editing yields the highest overall satisfaction rate, its difference from the second-best (anchor conditioning + SFT + naive editing) is not statistically significant (p ≈ 0.2). Overall, Table 1 result suggests that anchor conditioning during training contributes more significantly to task performance than the wBC objective. Importantly, our multi-step editing strategies consistently outperform best-of-N, demonstrating a key contribution of our approach in effectively leveraging inference-time computing to improve constraint satisfaction.
>
>    For the protein design task, where the goal is to discover maximally diverse solutions within a fixed sample size, both wBC and wBC + entropy significantly outperform SFT (p < .00001), although, the differences between inference strategies are not statistically significant. Regarding edit distance variations, the addition of entropy regularization to wBC appears to promote the exploration of diverse trajectories during inference. This results in a wider range of attempts to satisfy constraints, explaining the observed increase in average edit distances from the wild-type protein compared to wBC alone.
>
> 3. **Presentation and Technical Contribution**: We acknowledge that presenting a unified framework for two different domains with distinct evaluation setups is challenging. In proteins, we can create an edit between every pair of mutants, while in style transfer, each review has limited paraphrases. This necessitated a more sophisticated sampling strategy to ensure diverse edit pair coverage in the language domain. We appreciate your feedback on the presentation and will improve it in the revision. In the updated manuscript, we will add an algorithm box, provide more formal descriptions of our method, and better highlight the connections to weighted BC literature.

---

> > ### Author Response · Authors · 2024-10-27
> > **Continued Rebuttal for Reviewer VzBx**
> >
> > 4. **Clarity and MDP Formulation**: To address your concerns about clarity, we briefly describe the editing process as a Markov Decision Process:
> >
> >    The language model is the actor in the Markov Decision Process (MDP), that learns to navigate across a multi-attribute space defined by a set of attribute evaluators $C = \{c_1, c_2, ..., c_k\}$ (which can be classifier probability, regressor, embedding similarity, protein attribute predictors, etc). All attribute evaluators convert sequential inputs into a scalar value within a finite range ($c_j(.) \in [v_{j,min}, v_{j,max}]$). Each MDP episode begins with the initial state containing a context $x$ (that can be empty), a starting sequence $y_0$ and its attribute location $C(y_0)$ and a set of threshold window constraints $T = \{t_1, t_2, ..., t_k\}$, where $t_j = (t_{j, start}, t_{j, end})$ is the threshold boundary for attribute $c_j$. The rewriting language model $M$ iteratively edits the previous sequence until it satisfies the given threshold constraints, i.e., $P_M(y_{i+1}|x, y_i, C(y_i), T)$. $C(y_i)$ represents a vector of attribute scores for an intermediate output $y_i$. Here, each edit $y_i \rightarrow y_{i+1}$ is considered an action, with a deterministic transition to the next state. During inference, the goal is to generate a series of consecutive edits starting from $y_0$ to $y_n$, such that $C(y_n) \in T$.
> >
> >    However, to solve this problem with offline data only, we isolate the single editing step and only train on offline edit pairs. In the revised version, we'll include the MDP formulation and a pseudo-code to illustrate how MACS finetuning works starting from a collection of edit pairs.
> >
> > 5. **Addressing overall design concerns**
> >    MACS provides a coherent framework, demonstrating significant improvements over strong baselines in both text and protein domains. The core of MACS - creating edit pairs, training language models with SFT followed by wBC, and multi-step inference - is domain-agnostic and forms the backbone of our approach. The domain-specific modifications, such as kNN-sampling to deal with the sparsity of edit pairs in text and entropy regularization to increase the diversity of generated proteins, are introduced to tackle domain-specific problems and showcase the flexibility of our framework.
> >
> > We are grateful for the reviewer's thoughtful analysis, which has helped us better articulate the nuances of our method's performance across different domains. These insights will be invaluable as we continue to refine and expand our approach in future works. We look forward to incorporating these clarifications in our revised manuscript to provide a more comprehensive evaluation of our method's strengths and limitations.
> >
> > **References**:
> > [1] Fleiss, J. L., Levin, B., & Paik, M. C. (2013). Statistical methods for rates and proportions

---

### Review · Reviewer_p7fm · 2025-02-02

**Summary Of Contributions:**

The submission proposes to control the LLM output with satisfied properties/constraints in real value. The authors propose an attribute evaluator and an iterative inference method to rewrite the language model output. Further, the authors propose two benchmarks with multi-attribute constraints and evaluate their approach.

**Audience:**

Yes

**Claims And Evidence:**

Yes

**Requested Changes:**

1. More experiments are required.
2. Differences between this method and existing methods are required in detail.

**Strengths And Weaknesses:**

Strengths:
1. The proposed method is reasonable and the authors focus on the multi-attribute constraints, which is more challenge and important than previous works.
2. Several proposed techniques are interesting and meaningful, such as the Anchor Conditioned Inference and Multi-Step Reward Prioritized Inference.

Weaknesses:
1. The main challenge is the novelty of this work, though the authors criticize previous works, such as the iterative refine and inference method, the proposed method is actually in a same way with small modifications.
2. Besides, the results are not good as satisfied, even as the author claimed by themselves, some of the results are not significant. Also, the compared baselines are not enough, which is quite limited.
3. Further, though the overall framework is reasonable, the authors use RoBERT-large to train the evaluator. Since the evaluator is a regression-based training model, the regression-model would be not so precise as expected, this happens in all regression-based models. Therefore, the evaluator may not be so good. Though this is a hard problem, it should not be ignored.

---

> ### Author Response · Authors · 2025-02-14
> **Rebuttal for Reviewer p7fm**
>
> We sincerely thank the reviewer for their thorough and constructive feedback. We are particularly encouraged by their recognition of our work's ambitious focus on multi-attribute constraints as _"more challenging and important than previous works"_, and their acknowledgment of several of our technical contributions like Multi-Step Reward Prioritized Inference and Anchor Conditioned Inference as _"interesting and meaningful"_.
>
> 1. **Regarding concerns about the novelty of the work**
>     While we acknowledge building upon valuable prior work in iterative refinement [1,2], our contributions constitute significant advances through: (1) MACS - a comprehensive framework for fine-grained control across multiple continuous attributes simultaneously, featuring novel components like our generalized reward function for arbitrary scalar evaluators and reward-prioritized sampling for multi-step inference, (2) domain-specific technical innovations that showcase our framework's flexibility - k-NN sampling to address sparsity in text attribute space, anchor conditioning to maintain semantic consistency during editing, and entropy regularization to increase diversity in protein generation, and (3) the introduction of FineCS, a carefully designed benchmark with two challenging finegrained control tasks and rigorous evaluation protocols that will help standardize assessment of fine-grained multi-attribute control methods. Unlike previous approaches of extrapolative iterative refinement [1] and self-correction [2], that focus on unidirectional optimization of a single attribute, MACS provides the first unified framework that can work with any sequential domain, multiple arbitrary black-box evaluators, finegrained constraints, and domain-specific improvements while maintaining strong performance across diverse applications.
> 2. **Regarding the experimental results and baselines**
>     We respectfully note that our evaluation framework is more comprehensive than prior work in several ways. We carefully designed our experiments to ensure fair comparisons by giving all methods, including baselines, equivalent inference budgets. We also include best-of-N comparisons, which previous studies omitted [1, 2], and found it to be a strong baseline that is often time comparable and sometimes even outperforming the naive multi-step editing methods of previous studies. We demonstrate that using 8x smaller fine-tuned models, our method consistently outperforms in-context learning baselines with larger instruction-tuned Llama models. In Table 1 results, without anchor conditioning, MACS with weighted behavior cloning and reward prioritized sampling beats the other two inference strategies with statistical significance. However, this lead is diminished with the anchor conditioning setup as reported in the paper.
>     For protein design, unlike text style transfer where the goal is to find one satisfactory solution per instance, the task requires discovering maximal novel proteins satisfying multiple finegrained constraints outside the training data within a fixed inference budget. Even in this challenging setting, models trained with the MACS framework discover more unique and successful mutant sequences compared to strong baselines like Recombine and our strengthened Unique Recombine variant with statistical significance, demonstrating the framework's ability to generalize beyond text domains.
> 3. **Regarding the concern about using regression-based RoBERTa evaluator**
>     We acknowledge this thoughtful point. Our choice of regressors over classifiers was deliberate to maintain consistency across different attributes and evaluation setups. The framework's effectiveness does not depend on this specific choice - it can work with any black-box evaluator (for example, regressor or even protein folding softwares like Foldx), as long as its output is a real-value scalar. Our primary goal was to demonstrate that language model rewriting can effectively satisfy constraints from multiple arbitrary external evaluators. We plan to extend our framework to include categorical and lexical constraints in the future.
>
> Thank you again for helping us improve this work. We believe addressing these points strengthens the paper's contributions to advancing controllable text generation and protein design.
>
> **References:**
> [1] Padmakumar et. al. 2024, Extrapolative Controlled Sequence Generation via Iterative Refinement (https://arxiv.org/abs/2303.04562)
> [2] Welleck et. al. 2023, Generating Sequences by Learning to Self-Correct (https://openreview.net/pdf?id=hH36JeQZDaO)

---

### Decision · Action_Editor_vqTu · 2025-04-26

**Recommendation:** Accept as is

**Comment:**

I am a new AE reassigned to this paper as of today.

The reviewers praise a number of aspects of this work, including the multi-aspect rewriting setting and aspects of the methodology. I found the paper to be interesting and well-written.

The reviewers raise a number of criticisms, but I don't think these are critical for TMLR.

1. One of the main criticisms from reviewer p7fm is novelty, which is not in scope as a critique for TMLR.

2. Insufficient comparison to baselines is a more germane critique, which potentially impacts the supportedness of the claims. I partially agree that additional methods could be used.  For instance, FUDGE (Kevin Yang et al.) uses steering classifiers which I believe could be trained given the data in FineCS.  Moreover, the technical limitation that the method of Mireshghallah et al. can only use masked language models for generation was relaxed by follow-on work ( https://arxiv.org/pdf/2312.04510 ). However, these approaches don't precisely match the setup of the paper here. The approach here has certain simplicity advantages that I think make it fair for it to stand on its own without comparison to these.

3. I do think that JCoR's suggested baseline of self-prompting is also relevant. Small models are increasingly able to do this kind of refinement. However, as I understand the data condition and the models here, this baseline would not work in the protein design domain, as the authors point out in the response. I think it's fair to only consider approaches that cover both domains here.

4. VzBx's critiques about the lack of generality and ad hoc nature don't really resonate with me. Although aspects of the implementation like kNN sampling do strike me as ad hoc in a certain sense, the framework itself is general.

5. VzBx expresses concern about the small sizes of the gains in the results and some of the gains not being statistically significant, particularly as they relate to wBC over SFT.  Whether wBC is needed over SFT is not a critical part of the claimed approach in my view. Although it is part of the framework, wBC is not a novel algorithmic contribution of this work either; I view the use of SFT, wBC, or the combination as a bit like a hyperparameter choice, and it is not critical for a certain value of the hyperparameters to work better for the work to be accepted.

A few cites (Andrychowicz et al. on page 4, Peng et al. at the bottom of page 5) use citet when they should use citep.

**Audience:**

Yes

**Claims And Evidence:**

The claims of this papers are supported. The main claim I see (taken from the abstract) are:

> we create Multi-Attribute Constraint Satisfaction (MACS), a generalized method capable of finetuning language models on any sequential domain to satisfy user-specified constraints on multiple external real-value attributes.

This is true, and I see the generality across two domains as a key strength of the work.

> Our empirical results show that MACS achieves the highest threshold satisfaction in both FineCS tasks, outperforming strong domain-specific baselines.

Based on my interpretation of the results, this is true, although the improvements in some cases are not statistically significant.